# Increasing pediatric HIV testing positivity rates through focused testing in high-yield points of service in health facilities—Nigeria, 2016-2017

Solomon Odafe[1]*, Dennis Onotu[1], Johnson Omodele Fagbamigbe[1], Uzoma Ene[1], Emilia Rivadeneira[2], Deborah Carpenter[2], Austin I. Omoigberale[3], Yakubu Adamu[4], Ismail Lawal[4], Ezekiel James[5], Andrew T. Boyd[2], Emilio Dirlikov[2], Mahesh Swaminathan[1]

1 Division of Global HIV and Tuberculosis, Center for Global Health, Centers for Disease Control and Prevention, Abuja, Nigeria, 2 Division of Global HIV and Tuberculosis, Center for Global Health, Centers for Disease Control and Prevention, Atlanta, GA, United States of America, 3 Department of Pediatrics, University of Benin Teaching Hospital, Benin City, Nigeria, 4 Walter Reed Army Institute of Research–Military HIV Research Program, Abuja, Nigeria, 5 HIV/AIDS Care and Treatment, United States Agency for International Development, Washington, Dc, United States of America

* wsp7@cdc.gov

**Data Availability Statement:** All relevant data are within the manuscript and its Supporting Information files.

## Abstract

### Background

In 2017, UNAIDS estimated that 140,000 children aged 0–14 years are living with HIV in Nigeria, but only 35% have been diagnosed and are receiving antiretroviral therapy. Children are tested primarily in outpatient clinics, which show low HIV-positive rates. To demonstrate efficient facility-based HIV testing among children aged 0–14 years, we evaluated pediatric HIV-positivity rates in points of service in select health facilities in Nigeria.

### Methods

We conducted a retrospective analysis of HIV testing and case identification among children aged 0–14 years at all points of service at nine purposively sampled hospitals (November 2016–March 2017). Points of service included family index testing, pediatric outpatient department (POPD), tuberculosis (TB) clinics, immunization clinics, and pediatric inpatient ward. Eligibility for testing at POPD was done using a screening tool while all children with unknown status were eligible for HIV test at other points of service. The main outcome was HIV positivity rates stratified by the testing point of service and by age group. Predictors of an HIV-positive result were assessed using logistic regression. All analyses were done using Stata 15 statistical software.

### Results

Of 2,180 children seen at all facility points of service with unknown HIV status, 1,822 (83.6%) were tested for HIV, of whom 43 (2.4%) tested HIV positive. The numbers of children tested by age group were <1 years = 230 (12.6%); 1–4 years = 752 (41.3%); 5–9 years = 520 (28.5%); and 10–14 years = 320 (17.6%). The number of children tested by point of

**Funding:** This work was supported by the President's Emergency Plan for AIDS Relief (PEPFAR) through Centers for Disease Control and Prevention (CDC) under the terms of GH002097, GH002098, GH002099 and GH002100.

**Competing interests:** The authors have declared that no competing interests exist.

service were POPD = 906 (49.7%); family index testing = 693 (38.0%); pediatric inpatient ward = 192 (10.5%); immunization clinic = 16 (0.9%); and TB clinic = 15 (0.8%). HIV positivity rates by point of service were TB clinic = 6.7% (95% Confidence Interval (CI): 0.9–35.2%); pediatric inpatient ward = 4.7% (95%CI: 2.5–8.8%); family index testing = 3.5% (95%CI: 2.3–5.1%); POPD = 1.0% (95%CI: 0.5–1.9%); and immunization clinic = 0%. The percentage contribution to total HIV positive children found by point of services was: family index testing = 55.8% (95%CI: 40.9–69.8%); POPD = 20.9% (95%CI: 11.3–35.6%); inpatient ward = 20.9 (95%CI: 11.3–35.6%) and TB Clinic = 2.3% (95%CI: 0.3–14.8%). Compared with the POPD, the adjusted odds ratio (95% CI) for finding an HIV positive child by point of service were TB clinic = 7.2 (95% CI: 0.9–60.9); pediatric inpatient ward = 4.9 (95% CI: 1.9–12.8); and family index testing = 3.7 (95% CI: 1.5–8.8). HIV-positivity rates did not significantly differ by age group.

## Conclusion

In Nigeria, to improve facility-based HIV positivity rates among children aged 0–14 years, an increased focus on HIV testing among children seeking care in pediatric inpatient wards, through family index testing, and perhaps TB clinics is appropriate.

## Background

HIV/AIDS has significantly impacted the health of children globally since the beginning of the pandemic [1]. There are about 1.7 million children aged 14 years and below living with HIV worldwide in 2018, and about 54% of them are receiving lifesaving antiretroviral treatment (ART) [2]. In 2018, the Joint United Nations Programme on HIV/AIDS (UNAIDS) estimates that they were 140,000 children living with HIV in Nigeria [3]. However, compared with global achievements, there have been relatively slower progress in Nigeria with only 35% of HIV infected children receiving ART in 2018 [3].

Government of Nigeria (GON) first commenced the national HIV treatment program in 2002, and with support from the United States (US) government through US President's Emergency Plan for AIDS Relief (PEPFAR) has significantly expanded HIV care and treatment services in the country [4–6]. The initiation of people living with HIV (PLHIV) on ART in Nigeria was initially restricted to tertiary health centers due to weak health systems in other levels of care. In 2004, following rapid scale-up of ART services and infrastructural upgrades, secondary hospitals began providing ART services to children in Nigeria. However, pediatric HIV treatment coverage in Nigeria lags behind adult HIV treatment coverage [7]. To improve treatment coverage in children, GON adopted the use of family index testing as a key strategy for improving case finding in its accelerated plan for scaling up access to pediatric HIV treatment services between 2016 and 2018 [8]. However, progress with implementation has been slow. A trend analysis of national data indicated that pediatric ART coverage (based on CD4 cell count eligibility criterion of <350cells/μL) from 2010–2014 improved marginally from 10.2% to 20.7% while the adult ART program reached nearly half of all adults requiring ART in 2014 [7]. The UNAIDS 2018 estimates showed a disproportionately lower treatment coverage in children aged 0–14 years (35%) and in adult males aged 15 years and over (37%) compared with adult females aged 15 years and older (68%) [3]. Major reasons for the disproportionately lower coverage among children include among others, lower HIV testing

positivity rates, indicating poorly-targeted HIV testing, and lower linkage to treatment among children.

In 2016, PEPFAR adopted a program strategy geared towards working with GON and other partners to refocus efforts in a small number of prioritized high–burden Local Government Areas (LGAs), designated as "scale-up LGAs", to achieve HIV epidemic control by the end of the fiscal year 2018 (FY18). This is in line with the ambitious UNAIDS 90-90-90 targets of having 90 percent of PLHIV diagnosed, 90 percent of those diagnosed on ART, and 90 percent of those on ART virally suppressed by 2020. Achieving these targets also demands dramatic progress in closing the treatment gap for children [9], and this requires finding children living with HIV as a first step.

A blanket approach to HIV testing has not worked well for pediatric HIV case identification in HIV programs in Africa [10–12]. In Nigeria, blanket testing has not worked even in high-burden LGAs. As of June 2016, program monitoring results from PEPFAR-supported facilities in scale-up LGAs showed an overall HIV-positivity rate of 1.1% for children aged 0–14 years. The overall HIV-positivity rate for children and adolescents aged 0–19 years tested between October 2015 and September 2016 at all PEPFAR–supported LGAs was 0.8% with a 71% linkage to treatment rate for those who tested positive during this period. However, published reports from similar setting suggests that a more targeted approach to test may improve testing yield [13]. These statistics are indicative of a need for more effective targeted testing strategies in order to improve testing efficiency, by increasing testing yield, and efficacy, by linking more HIV-positive children to care and treatment.

To address the issues of low HIV testing yield and linkage among children, we piloted a pediatric intensive case-finding (PICF) approach at selected health facilities in Nigeria to identify efficient HIV testing points to identify children living with HIV for early ART initiation and to develop effective strategies for scale-up of pediatric HIV case finding and linkage in Nigeria.

## Methods

### Study design

This is a retrospective analysis of available program data collected during a PICF pilot in Nigeria.

### Study population

Data from all children aged less than 15 years seeking care in selected points of service (POS) in selected secondary level hospitals during the PICF pilot were included in the analysis.

### The major outcome measures

The HIV positivity rate among children tested, linkage rate to treatment among children tested positive, HIV results by age group and point of services, the proportion of total positive by point of service, and predictors of HIV positive result among children tested in pilot sites were the main outcome measures. Patients factors included in the model for odds ratio were age group and HIV testing point of service.

### Description of sites and process of site selection for pilot

In October 2016, a pediatric program gap analysis was conducted across ART sites in Nigeria by comparing the proportion of HIV positive children on ART with those of adults on ART. The review team looked at the total clients on ART at each site to determine the ratio of

children <15 on ART compared to that of adults currently on ART. It was expected that 6% of individuals on ART within each facility should be <15 years, based on the national proportion of children living with HIV among all PLHIV at the time of review (220,000 of 3,400,000 total estimated PLHIV) [14]. The pediatric ART gap for each site was calculated by estimating the additional number of children needed to be put on ART to make the ratio up to 6% of total clients currently on ART. The PICF pilot was conducted from November 15, 2016, to March 31, 2017, in selected treatment sites supported by PEPFAR in Nigeria. The selection was guided by the pediatric HIV program gap analysis. Nine secondary level hospitals with the widest gap were selected from the over four hundred treatment sites supported by PEPFAR to participate in the pilot.

## Strategies implemented during PICF pilot

Strategies for improving pediatric HIV case identification were implemented at all points of service (POS) for pediatric care seekers at the nine selected hospitals between November 15, 2016, to March 31, 2017. The POS were family index testing, pediatric outpatient department (POPD), tuberculosis (TB) clinics, immunization clinics, and pediatric inpatient ward. The family index testing POS identified new cases through a previously identified case, i.e. children of an HIV positive mother or siblings of an index child. HIV positive mothers were identified during counseling, at the ART clinic, ante-natal clinic, post-natal clinic or outpatient clinics and encouraged to return to the clinic with their children for HIV test. Additionally, parents of a previously identified HIV positive child were also encouraged to bring other children with unknown HIV status for HIV tests. At the TB clinic and pediatric inpatient ward, HIV testing was offered to all children whose parents reported that the child's HIV status was unknown. Targeted testing in immunization clinics was introduced, based on maternal HIV status; specifically, only infants whose mothers had unknown HIV status or were previously identified as HIV positive were eligible for testing. Finally, at the POPD children were screened to determine their eligibility for an HIV test using the Bandason screening tool [12].The Bandason screening tool consists of four questions [12]: "Has the child been admitted before?" "Does the child have recurring skin problems?" "Are one or both parents of the child deceased?" and "Has the child had poor health in the last 3 months?" An answer of yes to any of the four screening questions made the child eligible for an HIV test. The health workers at the selected sites were trained on the use of the Bandason screening tool. Advocacy about the PICF pilot was conducted to the leadership and staff of the selected sites and to the host state and local governments where these sites were located. These PICF strategies were piloted at the TB clinics, pediatric inpatient medical wards, adult outpatient clinics among families of identified index cases, POPDs, and the immunization clinics in the selected sites. All children less than 18 months of age were tested using HIV DNA PCR while children greater than 18 months of age were tested using a rapid diagnostic test, in accordance with World Health Organization testing guidelines [15]. The strategy for improving the linkage of newly-diagnosed to ART initiation was accompanied, immediate referral using volunteers to the ART clinic.

## Data analysis

The proportions of children tested and testing HIV positive were analyzed by age group, and percent contributions of each age group to total testing and total identified HIV-positive children were reported with 95% confidence intervals (CI). The proportions of children tested and testing HIV positive were then analyzed by POS. Additionally, the HIV-positivity rates from family index testing, POPD, inpatient ward, and TB clinics at pilot hospitals were reported with 95% CIs. To examine testing efficiency at the POS, we compared the contribution of each

POS to the total children tested and total identified HIV-positive children at the sites. To determine the linkage rate, we compared the number of children testing HIV positive at the POS of interest, with the number of HIV-positive children started on treatment from that POS. Furthermore, we calculated unadjusted and adjusted (adjusting for testing POS when comparing age group and for age group when comparing the testing POS) odds ratios for identification of pediatric HIV case by covariates of age group and POS. To account for the clustered design (i.e., design effect of the nine sites for combined analysis), we used a weighted analysis accounting for clustering by site. All analysis was done using Stata 15 (StataCorp. 2017. Stata Statistical Software: Release 15. College Station, TX: StataCorp LP). We accounted for the complex survey design (i.e., clustering and weighting) using svyset and svy procedures in Stata during data analysis

## Ethics statement

The study protocol was reviewed and approved by the National Health Research Ethics Committee of Nigeria (NHREC). The protocol received a non-research determination from the US Centers for Disease Control and Prevention (CDC) Atlanta. Patient informed consent was not required as only routine, anonymous, operational monitoring data were collected and analyzed.

## Results

### HIV testing outputs from all POS by age group

Information on outputs from the HIV testing service by age group from all POS from the nine pilot sites are summarized in Table 1. A total of 6,747 children who visited the participating secondary level sites during the pilot period were enumerated. Overall, a total of 2,180 (32.3%) of the enumerated children had unknown HIV status. Of those with unknown HIV status, 2,021 (92.7%) had parents or guardians who consented to have them tested for HIV, and among those with consent provided, 1,822 (90.2%) were tested for HIV. Among those tested, 43 (2.4%) of the children were found to be HIV positive. Children in the age group 1 to 4 years contributed the largest proportion (41.3%, [95% CI 39.0–43.6%]) of the total number of

**Table 1. HIV testing pediatric cascade by age group across all PICF pilot sites.**

| Indicator | Age Group in Years | | | | |
|---|---|---|---|---|---|
| | <1 year | 1–4 yrs. | 5–9 yrs. | 10–14 yrs. | Total |
| Number children enumerated (N) | 3,164 | 1,400 | 1,254 | 929 | 6,747 |
| Number children with Unknown HIV status and eligible for HIV test (n [col %]) | 382 (12.1%) | 851 (60.8%) | 575 (45.9%) | 372 (40.0%) | 2,180 (32.3%) |
| Number children with Unknown HIV status whose guardian consented for HIV test (n [col %]) | 332 (86.9%) | 833 (97.9%) | 521 (90.6%) | 335 (90.1%) | 2,021 (92.7%) |
| # Tested (n [col %]) | 230 (69.3%) | 752 (90.3%) | 520 (99.8%) | 320 (95.5%) | 1,822 (90.2%) |
| # Tested HIV+ (n [col %]) | 4 (1.7%) | 19 (2.5%) | 12 (2.3%) | 8 (2.5%) | 43 (2.4%) |
| # Initiated on ART (n [col %]) | 4 (100%) | 19 (100%) | 12 (100%) | 8 (100%) | 43 (100%) |
| Testing yield (%[95%CI]) | 1.7% (0.7–4.6) | 2.5% (1.6–3.9) | 2.3% (1.3–4.0) | 2.5% (1.3–4.9) | 2.4% (1.8–3.2) |
| Age group percent contribution to total HIV testing (%[95%CI]) | 12.6% (11.2–14.2) | 41.3% (39.0–43.6) | 28.5% (26.5–30.7) | 17.6% (15.9–19.4) | 100% |
| Age group percent contribution to total HIV+ | 9.3% (3.5–22.3) | 44.2% (30.2–59.1) | 27.9% (16.6–43.0) | 18.6% (9.6–33.0) | 100% |

children tested, followed by children in the age group 5 to 9 years (28.5%,[26.5–30.7]). Similarly, children in the age groups 1 to 4 years and 5 to 9 years contributed 44.2% (30.2–59.1%) and 27.9% (16.6–43.0%) of the total number of children that tested HIV positive. Of note, all 43 HIV-positive children identified were linked to ART initiation.

## HIV testing acceptance for all age groups by POS

The results from the HIV testing service by POS for all age groups and all pilot sites are summarized in Table 2. Of the 2,180 children with unknown HIV status seen at all POS, parents or guardians of 2,021 children (92.7%) accepted that their children be tested for HIV. In family index testing, 807 out of 890 parents (90.7%) gave consent for their children to be tested. Acceptance rates for HIV testing was 76.4%, 95.3%, 99.5% and 100% for immunization clinic, POPD, pediatric inpatient ward and TB clinic respectively.

## HIV testing outputs from all sites by POS

HIV positivity rates by point of service were TB clinic = 6.7% (95% Confidence Interval (CI): 0.9–35.2%); pediatric inpatient ward = 4.7% (95%CI: 2.5–8.8%); family index testing = 3.5% (95%CI: 2.3–5.1%); POPD = 1.0% (95%CI: 0.5–1.9%); and immunization clinic = 0%. The largest proportion of children tested were from the POPD (49.7% [95%CI: 47.4–52.0%]), followed by family index testing POS (38.0% [35.8–40.3%]). However, the largest proportion of the total number of HIV-positive children identified were from the family index testing POS (55.8% [95%CI: 40.9–69.8%]), followed by POPD (20.9% [95%CI: 11.3–35.6%]) and the inpatient ward, accounting for 20.9% (95%CI: 11.3–35.6%) of the total number of children testing HIV positive. The TB clinic POS had the highest yield, at 6.7% (0.9–35.2%), but contributed only 2.3% (0.3–14.8%) of the total number of children testing HIV positive.

In examining odds ratios of pediatric HIV case detection by age group, after adjusting for POS, using age group <1 year of age as the control, there was no significant difference in adjusted odds of HIV positivity rate by age group (Table 3). However, in examining odds ratios

**Table 2. HIV testing pediatric cascade by points of service across all PICF pilot sites.**

| Indicator | Points of service | | | | | |
|---|---|---|---|---|---|---|
| | Family index modality | POPD | TB clinic | Immunization clinic | Inpatient ward | Total |
| **Number children enumerated (N)** | 2,509 | 1,098 | 44 | 2,835 | 257 | 6,743* |
| **Number children with Unknown HIV status and eligible for HIV test (n [col %])** | 890 (35.5%) | 955 (87.0%) | 15 (34.1%) | 127 (4.5%) | 193 (75.1%) | 2,180 (32.3%) |
| **Number children with Unknown HIV status whose guardian consented for HIV test (n [col %])** | 807 (90.7%) | 910 (95.3%) | 15 (100%) | 97 (76.4%) | 192(99.5%) | 2,021 (92.7%) |
| **# Tested (n [col %])** | 693 (85.9%) | 906 (99.6%) | 15 (100%) | 16 (16.0%) | 192 (100%) | 1,822 (90.2%) |
| **# Tested HIV+ (n [col %])** | 24 (3.5%) | 9 (1.0%) | 1 (6.7%) | 0 (0%) | 9 (4.7%) | 43 (2.4%) |
| **# Initiated on ART (n [col %])** | 24 (100%) | 9 (100%) | 1 (100%) | 0 (0%) | 9 (100%) | 43 (100%) |
| **Testing yield (%[95%CI])** | 3.5% (2.3–5.1) | 1.0% (0.5–1.9) | 6.7% (0.9–35.2) | 0 (0%) | 4.7% (2.5–8.8) | 2.4% (1.8–3.2) |
| **Point of service percent contribution to total HIV testing (% [95%CI])** | 38.0% (35.8–40.3) | 49.7% (47.4–52.0) | 0.8% (0.27–3.1) | 0.9% (0.5–1.4) | 10.5% (9.2–12.0) | 100% |
| **Point of service percent contribution to total HIV+ (%[95%CI])** | 55.8% (40.9–69.8) | 20.9% (11.3–35.6) | 2.3% (0.3–14.8) | 0 (0%) | 20.9% (11.3–35.6) | 100% |

*Four patients enumerated in malnutrition clinic but not tested were not included in the table

**Table 3. Unadjusted and adjusted odds ratios of pediatric HIV case detection by patient age group and POS.**

| Patient Characteristics | Unadjusted Odds ratio (95%CI) | Adjusted Odds ratio (95%CI) |
|---|---|---|
| Age group | | |
| <1 year | 1 | 1 |
| 1–4 years | 1.5 (0.5–4.3) | 1.2 (0.3–4.1) |
| 5–9 years | 1.3 (0.4–4.2) | 1.1 (0.3–4.3) |
| 10–14 years | 1.4 (0.4–4.9) | 0.9 (0.2–4.4) |
| POS | | |
| POPD | 1 | 1 |
| Family index | 3.6 (1.7–7.7) | 3.7 (1.5–8.8) |
| Inpatient ward | 4.9 (1.9–12.5) | 4.9 (1.9–12.8) |
| TB clinic | 7.1 (0.8–60.2) | 7.2 (0.9–60.9) |

of pediatric HIV case detection by POS, after adjusting for age, compared with the POPD, the inpatient ward (Adjusted Odds Ratio (AOR) 4.9 (95% CI: 1.9–12.8)) and family index testing POS (AOR: 3.7 (95% CI: 1.5–8.8)) had significantly higher odds for finding a HIV positive child (Table 3). The AOR for TB clinic did not achieve statistical significance, but the magnitude of the effect was large and probably statistical significance was not achieved due to the small number of patients in the TB clinic.

## Discussion

Our study found that the pediatric inpatient wards and family index testing POS provided the highest number of HIV-positive children, while still maintaining high HIV positivity rates compared with other POS. The pediatric inpatient ward and family index testing modality also contributed a disproportionately higher amount of total identified HIV-positive children than their contributions to the total HIV testing, a fact supported by significantly higher odds ratios of finding HIV-positive children in these POS compared with that of the POPD. While there were as many children living with HIV identified in the POPD as in the inpatient ward, this result was coupled with a significantly lower yield. Even with the use of the HIV risk screening tool in the POPD, this POS gave a low yield of 1.0%. TB clinics had the highest yield, but TB clinics had fewer absolute numbers of pediatric HIV cases found because there were few pediatric clients seen in that POS. These findings suggest that a focus on pediatric inpatient and family index testing POS as pediatric HIV testing points might be the most efficient approach for HIV testing among children in Nigeria.

The entry point for HIV testing in children influences the proportion of cases found [16–18]. A study by Wagner et al. conducted in Nairobi, Kenya, where HIV-infected adults were encouraged to bring their children for HIV testing (family index testing), found that the rate of pediatric HIV testing increased 3.8-fold from 3.5% to 13.6% [16]. A similar study by Lewis Kulzer et al. conducted in the Nyanza province in Kenya reported that HIV testing through the family index approach increased case detection among children[17]. Both study findings suggest that scaling up family index testing has the potential of improving HIV case detection among children. We also found that the family index testing modality was the most efficient POS for all sites, accounting for about 38% of total children tested but contributing more than half (55.8%) of all the positives identified.

One of the key recommendations of the national acceleration plan for improving pediatric case finding in Nigeria was the use of family index testing[8]. The findings from our study give credence to the recommended approach in the national acceleration plan. Furthermore, a study conducted in Cameroon that compared case detection in biological children of HIV-

positive parents (targeted testing) and other children at the POPD (blanket testing), reported a higher HIV positivity rate of 3.5% in the targeted group compared with 1.6% in the blanket group[10].The HIV positivity rate of 3.5% among children identified through family index testing in our study was similar to the HIV-positive rate found in biological children of HIV-positive parents in the study in Cameroon[10].

Our study found a relatively high HIV positivity yield of 4.7% among children in the inpatient ward. A study conducted in Zambia among hospitalized children reported a high HIV positivity rate of 29.2% and concluded that the inpatient ward has a huge potential for identifying children with HIV[19]. The differences in HIV positivity rates in our study and the Zambian study was the population HIV prevalence. Whereas Nigeria's adult population HIV prevalence was 1.5% [14] that for Zambia was 11.3% [20]. Nonetheless, there was a consistently high HIV case finding among children in the inpatient ward in both studies.

PEPFAR program guidance suggests that testing children not only in pediatric wards, and but also in POPDs among children screening as high risk may increase the case detection of HIV-infected children [21]. Our findings suggest that the pediatric inpatient ward was an efficient POS for finding children living with HIV. Although the contribution of inpatient (20.9%) to case finding was similar to POPD (20.9%), a lot more tests had to be done at POPD to find the same number of patients as the inpatient ward. The observed HIV positivity yield of 1% at POPD in our study using the Bandason tool was much lower than the reported HIV positivity yield of 4% at POPD by Yumo et al using a symptoms-based testing strategy in Cameroon [10]. The differences in HIV positivity rates were due to differences in general population HIV prevalence, the adult HIV prevalence in Cameroon is much higher, 3.6% compared with 1.5% in Nigeria [3, 22, 23]. In both studies, the HIV test results at POPD was similar to the population HIV prevalence. However, considering the relatively large number of children that needed to be tested to find a positive case at the POPD despite the use of a screening tool in our study, there is a need to re-assess the validity of the Bandason screening tool in low HIV prevalence settings like Nigeria and among children, less than six years since the Bandason tool was only tested for children above six years [12].

Additionally, we found the highest HIV positivity rate (6.7%) and the highest adjusted odds ratio for finding a positive result among children at the TB clinic. In a much larger study among children infected with TB conducted by Tilahun et al in Ethiopia, 291 children were tested for HIV and 28.2% of them were HIV positive [24]. Our findings suggest all children in TB clinic should be tested for HIV, but TB clinic cannot be counted on to provide high absolute numbers of children living with HIV

In our study we observed that all the children identified through the POS were also initiated on treatment, giving a linkage to treatment rate of 100%. The sites employed a strategy of accompanied referral, which used volunteers to immediately accompany parents or guardians and newly diagnosed children to ART POS to ensure all children identified HIV positive were linked to treatment. The approach assisted parents/guardians to quickly navigate the ART clinic and obtain care for their children. The excellent linkage rate observed in our study was comparable to those reported for a South African ART program that utilized escort services to facilitate linkage to treatment among HIV positive adults, and scale-up of this approach in the Nigeria HIV program will ensure that identified HIV-positive children are offered ART [25].

We found HIV testing acceptance rates of 76.4%, 90.7%, 95.3%, 99.5% and 100% for parents/guardians at immunization clinic, family index testing, POPD, inpatient ward and TB clinic respectively. A systematic review of several studies conducted mainly in Kenya and Uganda by Govindasamy et al reported the highest acceptance of HIV testing (86.3%) by parents/guardians at inpatient wards and lowest acceptance (51.7%) in the family index modality [26]. Compared with reports by Govindasamy et al, with the exception of the

immunization clinic, our study found a much higher HIV testing acceptance rate among all the provider-initiated testing and counseling POS. The finding suggests that the pre-test counseling provided for families of at-risk pediatric patients at our study sites was effective. Furthermore, a study conducted in Cameroon by Yumo et al reported high HIV testing acceptance rates for children among both HIV-positive parents (99.7%) and other guardians/parents at POPD (98.8%). In our study, HIV acceptance rates were comparable to those reported by Yumo et al for Cameroon [10].

However, even with this high acceptance by guardians to have at-risk pediatric patients tested for HIV, we found that there were missed opportunities for testing at the immunization clinic and at the family index testing POS. We found that 76.4% of mothers accepted the HIV test for their babies but only 16.5% of children with unknown status at the immunization clinic were eventually tested for HIV. A systematic review examining HIV testing in immunization clinics in children reported an acceptance rate of 89.5 to 100%. However, in that systematic review, only about 56.8% to 86.0% of the children were eventually tested [27]. While the acceptance of HIV testing in the immunization clinic in our study was only slightly lower than that in the systematic review, the percentage of children tested at immunization clinics in our study was much lower than those previously reported. While previous studies have reported that immunization clinic may be a promising area for identifying and testing HIV exposed children in high HIV prevalence settings [28, 29],the low HIV positivity rate among children tested at the immunization clinics in our study suggests that scaling up testing at immunization clinic as it is currently done may not be an efficient way to find pediatric HIV cases.

There were also missed opportunities at the family index testing POS. About 24% of children with unknown status identified through the family index POS were not tested for HIV. However, the percentage of children with unknown status that was not tested (24%) in our study was much lower than the 43.3% reported by Yumo et al for neighboring Cameroon[30]. A major reason for the missed opportunity was that at this POS, the method for getting these children tested was simply to ask parents to return to the clinic with the children, and parents often did not return with their children for HIV testing as scheduled. A similar factor played a role in the high missed opportunity in the study in Cameroun [10]. The researchers insinuated that the reason for the high missed opportunity in their study was because parents initially came for their own care and did not have their children with them at the time of accepting to test their children and eventually failed to follow through [10]. Our study did not investigate the reasons for the failure of parents or guardians to return with their children. However, a previous study in Nigeria found that HIV infection among family income earners adversely affects families socioeconomically and impacts on parents' ability to continue to seek health services for their children, including for family index testing [28]. This suggests that a home-based HIV testing and counseling approach for children of consenting parents/guardian, or health outreach workers going to the home to conduct testing of children, may be required for the successful implementation of the family index testing.

In examining the pediatric HIV testing cascade by age group in our study, it is noteworthy that although the age group <1 year had the most children accessing care, it also had the lowest proportion of children with unknown HIV status, and lowest percentage testing HIV positive. This may be because all sites involved in our study had a PMTCT program. Studies have demonstrated that PMTCT programs increase HIV test rates during ante-natal care in pregnant women, promote uptake of antiretroviral drugs during pregnancy and promote HIV testing among HIV exposed infants [31–34]. We found that the percentage of contributions by age group to total children tested were closely aligned to the proportions contributed to total HIV positive found. This finding was further supported by no statistically significant difference in odds ratios for finding an HIV-positive child by age group. Thus, our findings do not support

program emphasis on a specific age group. That said, because of a greater proportion of children with an unknown HIV status among 1-4-year-olds, the largest number of HIV cases were found among this age group. This indicates the need for health providers to consider HIV risk stratification for all children irrespective of age if no HIV status is documented when they present at a health facility.

The strength of our study is the data from a large number of samples of children analyzed in the study. However, the purposive nature of site selection limits the generability of the findings from our analysis. Additionally, other POS with the potential to give high positivity yields, such as malnutrition clinics, were not thoroughly assessed in our study. Nonetheless, the findings provide important insights into how HIV positivity rates and testing efficiency can be improved by focusing on testing and ensuring linkage, in targeted POS in health facilities.

## Conclusion

The pilot of pediatric intensified case finding demonstrated that the family index testing POS, followed by pediatric inpatient testing, was the most efficient testing streams for HIV testing among children. Specifically, these modalities identified high numbers of children living with HIV while maintaining high HIV positivity rates. Furthermore, while the POPD was an important testing point for identifying high numbers of HIV positive children, yield was poor, even with the application of a validated screening tool for identifying children at risk for having HIV. Testing among children attending TB clinic gives high yield but low absolute numbers. The study findings suggested that to improve facility-based HIV positivity rates among children aged 0–14 years in Nigeria, an increased focus on offering and ensuring HIV testing through family index testing, offering testing to all children in pediatric inpatient wards who do not know status, testing in POPD only if targeted using a validated screening tool tailored to the population, and among all children attending TB clinics is appropriate. This focus, coupled with the continuation of the 100% linkage to ART initiation among these children in the study, will allow Nigeria to make progress in closing the pediatric ART treatment gap.

## Supporting information

**S1 Data.**
(XLS)

## Acknowledgments

**Disclaimer:** The findings and conclusions in this manuscript are those of the authors and do not necessarily represent the views of the United States (U.S.) Centers for Disease Control and Prevention. The use of trade names is for identification only and does not imply endorsement by the U.S. Centers for Disease Control and Prevention or the U.S. Department of Health and Human Services.

## Author Contributions

**Conceptualization:** Solomon Odafe, Dennis Onotu, Johnson Omodele Fagbamigbe, Uzoma Ene, Yakubu Adamu, Mahesh Swaminathan.

**Data curation:** Solomon Odafe, Dennis Onotu, Johnson Omodele Fagbamigbe, Uzoma Ene, Emilia Rivadeneira, Deborah Carpenter, Ismail Lawal, Ezekiel James, Andrew T. Boyd, Emilio Dirlikov, Mahesh Swaminathan.

**Formal analysis:** Solomon Odafe, Dennis Onotu, Johnson Omodele Fagbamigbe, Uzoma Ene, Deborah Carpenter, Yakubu Adamu, Ismail Lawal, Ezekiel James, Emilio Dirlikov, Mahesh Swaminathan.

**Funding acquisition:** Solomon Odafe, Dennis Onotu, Emilia Rivadeneira, Deborah Carpenter, Yakubu Adamu, Ismail Lawal, Ezekiel James, Mahesh Swaminathan.

**Investigation:** Solomon Odafe, Dennis Onotu, Johnson Omodele Fagbamigbe, Uzoma Ene, Emilia Rivadeneira, Deborah Carpenter, Austin I. Omoigberale, Ezekiel James, Andrew T. Boyd, Emilio Dirlikov, Mahesh Swaminathan.

**Methodology:** Solomon Odafe, Dennis Onotu, Uzoma Ene, Emilia Rivadeneira, Deborah Carpenter, Austin I. Omoigberale, Andrew T. Boyd, Mahesh Swaminathan.

**Project administration:** Solomon Odafe, Johnson Omodele Fagbamigbe, Mahesh Swaminathan.

**Resources:** Solomon Odafe, Emilia Rivadeneira, Mahesh Swaminathan.

**Software:** Solomon Odafe, Mahesh Swaminathan.

**Supervision:** Solomon Odafe, Dennis Onotu, Johnson Omodele Fagbamigbe, Uzoma Ene, Emilia Rivadeneira, Deborah Carpenter, Austin I. Omoigberale, Yakubu Adamu, Ismail Lawal, Ezekiel James, Andrew T. Boyd, Emilio Dirlikov, Mahesh Swaminathan.

**Validation:** Solomon Odafe, Dennis Onotu, Johnson Omodele Fagbamigbe, Uzoma Ene, Emilia Rivadeneira, Deborah Carpenter, Austin I. Omoigberale, Yakubu Adamu, Ismail Lawal, Ezekiel James, Andrew T. Boyd, Emilio Dirlikov, Mahesh Swaminathan.

**Visualization:** Solomon Odafe, Dennis Onotu, Johnson Omodele Fagbamigbe, Emilia Rivadeneira, Deborah Carpenter, Austin I. Omoigberale, Yakubu Adamu, Ismail Lawal, Ezekiel James, Andrew T. Boyd, Emilio Dirlikov.

**Writing – original draft:** Solomon Odafe.

**Writing – review & editing:** Solomon Odafe, Dennis Onotu, Johnson Omodele Fagbamigbe, Uzoma Ene, Emilia Rivadeneira, Deborah Carpenter, Austin I. Omoigberale, Yakubu Adamu, Ismail Lawal, Ezekiel James, Andrew T. Boyd, Emilio Dirlikov, Mahesh Swaminathan.

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
