## [Decision Letter · Decision Letter 0]

24 Dec 2019

PONE-D-19-27505

Increasing pediatric HIV testing positivity rates through focused testing in high-yield facility-based settings—Nigeria, 2016-2017

PLOS ONE

Dear Solomon Odafe,

Thank you for submitting your manuscript to PLOS ONE. After careful consideration, we feel that it has merit but does not fully meet PLOS ONE’s publication criteria as it currently stands. Therefore, we invite you to submit a revised version of the manuscript that addresses the points raised during the review process.

We would appreciate receiving your revised manuscript by Feb 07 2020 11:59PM. To enhance the reproducibility of your results, we recommend that if applicable you deposit your laboratory protocols in protocols.io, where a protocol can be assigned its own identifier (DOI) such that it can be cited independently in the future. For instructions see: http://journals.plos.org/plosone/s/submission-guidelines#loc-laboratory-protocols

We look forward to receiving your revised manuscript.

Kind regards,

Joseph Fokam, Ph.D

Academic Editor

PLOS ONE

Journal Requirements:

Reviewers' comments:

Reviewer's Responses to Questions

**Comments to the Author**

1. Is the manuscript technically sound, and do the data support the conclusions?

Reviewer #1: Yes

Reviewer #2: Yes

Reviewer #3: Partly

2. Has the statistical analysis been performed appropriately and rigorously? 

Reviewer #1: Yes

Reviewer #2: N/A

Reviewer #3: Yes

3. Have the authors made all data underlying the findings in their manuscript fully available?

Reviewer #1: Yes

Reviewer #2: Yes

Reviewer #3: No

4. Is the manuscript presented in an intelligible fashion and written in standard English?

Reviewer #1: Yes

Reviewer #2: Yes

Reviewer #3: Yes

5. Review Comments to the Author

Reviewer #1: I deeply appreciated the opportunity to review this excellent work. It represents a great effort of the government of Nigeria to achieve the ambitious UNAIDS target of 90-90-90 by 2020. Table 3 is describing unadjusted and adjusted odd ratios of pediatric HIV case detection by patient age group and HIV positive results. However there is no p-value calculated for each variable (Age group and HIV positive results).

Reviewer #2: This paper is about identifying an efficient case-finding approach for children with HIV and linking them to care in Nigeria. It was conducted well and tells a good story about where to focus HIV testing of children in large hospital settings in Nigeria for efficient case-finding. Table 2 is particularly illuminating, it is easy to see where to target testing based on this table.

Overall the information is sound, but can benefit from some re-structuring of information for consistency and flow, particularly in the results and discussion and overall to make sure the abstract mirrors the paper’s findings and the methods, results, and discussion tie together well. See specific comments and suggestions below.

Abstract

Results are largely presented well, the results first provide a break down of the numbers test by age group, then by department, and then by positivity rate, in order of highest % to lowest.

Minor issue:

Results: In the number tested by age group 5-9 years=520 (28.5%) should follow the 1-4 year age group for consistency and flow.

Major issue:

Conclusion: Unclear why the recommendation ordering for focused testing is “pediatric inpatient wards, through family testing, and perhaps TB clinic is appropriate”. It is not based on positivity alone, therefore authors should indicate why those modalities (similar to manuscript conclusion)

Background,page 4

Minor issue:

There is redundancy in the last few sentences of the last paragraph, suggest streamlining.

Methods

Study design, page 4

Minor issue:

Suggest adding retrospective to the design.

Study population, page 4

Minor issue:

Regional information missing. Rural, urban, semi-urban?

Outcomes measures, page 4

Major issues:

Incomplete, what about number and proportion tested and positive in each department?

For factors associated, give examples (i.e. testing acceptance)

Description and process of site selection for pilot, page 4

Minor issues:

Suggest starting with the first research activity done, for instance if the gap analysis was first, start there and flow forward to details of determining children ration.

Cite (220,000/3,400,000 estimated PLHIV)…

Strategies, page 4

Nice overall description of strategies and process to roll out.

Minor issues:

Consider adding when strategies were implemented to the first sentence

What department(s) was family testing conducted in?

Data Analysis, page 5

Major issue

Section on data collection

In the Analysis section, suggest putting the sentence about STATA at the end

Results

Major issues – editing and additional information

HIV Testing Outputs from all sites by POS by age group, page 6

Linkage mention seems out of place in this paragraph, consider its own paragraph and placing it with HIV testing cascade flow.

HIV Testing Outputs from all sites by POS, page 6

This text draws from Table 2, which is a nicely presented table of findings. The extra details on the family testing may not be essential here, so evident in the table and other departments are not presented similarly.

Some redundancy of results mid-paragraph. Suggest refining and keeping any interpretation to Discussion.

Suggest that the results include a paragraph summarizes testing acceptance (based on Table 2), since it is discussed in the Discussion.

Discussion, page 7

Major issues - Some rewriting and reorganization is warranted:

Important points are captured here but suggested restructuring for better flow and presentation. For instance, first primary findings, then supporting / conflicting evidence, other findings, and limitations.

Ordering suggestions

Paragraph 1 - Suggest starting with overall primary findings that answers research question

Paragraph 2 – Suggest moving this further down since not about primary findings. Suggest adding references to support ideas here.

Paragraph 3 – Suggest move this paragraph up since it supports primary findings. Any references to substantiate findings on inpatient care? For TB consider adding information to validate Conclusion statement.

Paragraph 4 – Suggest keeping it after the above paragraph since tied to primary findings.

Paragraph 5 – fine, no comments

Paragraph 6 – Systematic review on acceptance, interesting but need to mention acceptance in results Suggest starting with what the study found (to put it in context), then introduce the systematic review and corroborating findings

Paragraph 7 – fine, no comments

Paragraph 8 – PEPFAR/inpatient ward– suggest weaving into primary findings, since it is provides support for inpatient findings.

Paragraph 9 – fine, no comments

Conclusion

Question about this sentence: “Specifically, these modalities identified high numbers of children living with HIV while maintaining high yields”

isn’t identifying high numbers of children living with HIV yield? Please clarify.

Reviewer #3: Based on presented data, some conclusions are not drawn appropriately. This issue is more pronounced in the discussion section. Please, see the review attachment for our comments in the different sections of the manuscript.

6. PLOS authors have the option to publish the peer review history of their article (what does this mean?). If published, this will include your full peer review and any attached files.

Reviewer #1: No

Reviewer #2: No

Reviewer #3: Yes: Habakkuk A. Yumo

---

## [Author Response · Author response to Decision Letter 0]

6 Feb 2020

Response to Reviewers’ Comments

Reviewer 1 Comments and Authors’ responses:

Reviewer #1: I deeply appreciated the opportunity to review this excellent work. It represents a great effort of the government of Nigeria to achieve the ambitious UNAIDS target of 90-90-90 by 2020. Table 3 is describing unadjusted and adjusted odd ratios of pediatric HIV case detection by patient age group and HIV positive results. However, there is no p-value calculated for each variable (Age group and HIV positive results).

Authors’ response:

We did not include the p values for the values as rightly observed by reviewer. However, we provided the 95% Confidence Intervals (CIs) for all the values presented. The determination statistical significance or Non-significance can be accurately made by using the 95% CIs[1].

Reviewer 2 Comments and Authors’ responses:

Overall the information is sound, but can benefit from some re-structuring of information for consistency and flow, particularly in the results and discussion and overall to make sure the abstract mirrors the paper’s findings and the methods, results, and discussion tie together well. See specific comments and suggestions below.

Authors’ response:

We have re-structured information in both the abstract, results and discussions to improve the flow and consistency throughout the manuscript. The information in the abstract also mirrors the content of the paper.

Abstract

Results are largely presented well, the results first provide a break down of the numbers test by age group, then by department, and then by positivity rate, in order of highest % to lowest.

Authors’ response:

We have now included reorganized the order of presentation of results by age group to improve the flow. Please see lines page 1, 29 to 31 of marked up version of revised manuscript.

Minor issue:

Results: In the number tested by age group 5-9 years=520 (28.5%) should follow the 1-4 year age group for consistency and flow.

Authors’ response:

Authors agree with reviewer’s suggestions. We have now included reorganized the order of presentation of results by age group to improve the flow. Please see page 1, lines 29 to 31 of marked up version of revised manuscript.

Major issue:

Conclusion: Unclear why the recommendation ordering for focused testing is “pediatric inpatient wards, through family testing, and perhaps TB clinic is appropriate”. It is not based on positivity alone, therefore authors should indicate why those modalities (similar to manuscript conclusion)

Authors’ response:

The POPD was not an efficient point of service: low yield, and two other points of service here (peds inpatient, and family index) had statistically significant odds ratio (OR) of finding HIV+ child than POPD. We mention TB clinic as “perhaps” because the OR did not reach statistical significance, but magnitude of effect was large and probably statistical significance was not achieved due to small number in TB clinic. Inclusion of POPD would seem to muddy these findings, not clarify them. We agree with the reviewer that this conclusion should not be based on positivity alone, but it isn’t. The odds ratio allows us to consider both positivity/yield and volume. We have included a comments on statistical significance of the AOR for TB clinic on page 7, lines 220 and 221 of marked up version revised manuscript.

Background,page 4

Minor issue:

There is redundancy in the last few sentences of the last paragraph, suggest streamlining.

Authors’ response:

The redundant sentences have been deleted, please see page 4, lines 95 to 97 of marked up version of revised manuscript

Methods

Study design, page 4

Minor issue:

Suggest adding retrospective to the design.

Authors’ response:

We have updated study design as recommended. Please see page 4, lines 100 and 101 of marked up version of revised manuscript.

Study population, page 4

Minor issue:

Regional information missing. Rural, urban, semi-urban?

Authors’ response:

The regional information on sites will not accurately define our population in this instance because the patients accessing care come from far and near to access services in the sites included for the study. Unfortunately, information on location of patients was not available for the analysis.

Outcomes measures, page 4

Major issues:

Incomplete, what about number and proportion tested and positive in each department?

For factors associated, give examples (i.e. testing acceptance)

Authors’ response:

We have reorganized the section on outcome measures. The patient’s factors included in the model for odds ratio were age group and HIV testing point of service. Please see page 4, lines 106 to 110 of marked up version of revised manuscript.

Description and process of site selection for pilot, page 4

Authors’ response:

We described the process of site selection in detail. Please see page 4, lines 112 to 123 of marked up version of revised manuscript.

Minor issues:

Suggest starting with the first research activity done, for instance if the gap analysis was first, start there and flow forward to details of determining children ration.

Cite (220,000/3,400,000 estimated PLHIV)…

Authors’ response:

We have reorganized the section based on reviewers recommendations. Please see page 4, lines 112 to 123 of marked up version of revised manuscript.

Strategies, page 4

Nice overall description of strategies and process to roll out.

Minor issues:

Consider adding when strategies were implemented to the first sentence

What department(s) was family testing conducted in?

Authors’ response:

We have reorganized the section based on reviewers recommendations. We have added dates the strategies were implemented and described the departments that family index testing was implemented Please see page 5, lines 138 to 145 of marked up version of revised manuscript.

Data Analysis, page 5

Major issue

Section on data collection

In the Analysis section, suggest putting the sentence about STATA at the end

Authors’ response:

We have reorganized the section based on reviewers recommendations. Please see page 6, lines 176 to 177 of the revised manuscript.

Results

Major issues – editing and additional information

HIV Testing Outputs from all sites by POS by age group, page 6

Linkage mention seems out of place in this paragraph, consider its own paragraph and placing it with HIV testing cascade flow.

Authors’ response:

In this section we described results presented on table 1 which included number of children initiated on ART. Therefore, the mention of linkage rate here was to ensure completeness of that information.

HIV Testing Outputs from all sites by POS, page 6

This text draws from Table 2, which is a nicely presented table of findings. The extra details on the family testing may not be essential here, so evident in the table and other departments are not presented similarly.

Authors’ response:

In this study the family Index Testing was treated as a POS. We have included a detailed description of the family index testing modality in the method section. See excerpt included in section on strategies implemented during PICF pilot of marked up version of revised manuscript (page 5, lines 138 to 144) 

Some redundancy of results mid-paragraph. Suggest refining and keeping any interpretation to 

Authors’ response:

We have refined section in line with authors comments.

Discussion.

Suggest that the results include a paragraph summarizes testing acceptance (based on Table 2), since it is discussed in the Discussion.

Authors’ response:

We have included a paragraph summarizing the acceptance rates on page 7, lines 198 to 204 of the result section of marked up version of revised manuscript.

Discussion, page 7

Major issues - Some rewriting and reorganization is warranted:

Important points are captured here but suggested restructuring for better flow and presentation. For instance, first primary findings, then supporting / conflicting evidence, other findings, and limitations.

Authors’ response:

We have reorganized the entire discussion session to improve flow.

Ordering suggestions

Paragraph 1 - Suggest starting with overall primary findings that answers research question

Authors’ response:

We have started with paragraph that summarized findings from the study, please see page 8, lines 243 to 255 of marked up version of revised manuscript.

Paragraph 2 – Suggest moving this further down since not about primary findings. Suggest adding references to support ideas here.

Authors’ response:

We have included references to support our ideas and also reorganized the paragraph. The paragraph as been moved further down and its now the new paragraph 8. See page 9, lines 295 to 300

Paragraph 3 – Suggest move this paragraph up since it supports primary findings. Any references to substantiate findings on inpatient care? For TB consider adding information to validate Conclusion statement.

Authors’ response:

The paragraph has been moved up, its now the new paragraph 1 in revised manuscript. 

Paragraph 4 – Suggest keeping it after the above paragraph since tied to primary findings.

Authors’ response:

We have maintained paragraph as suggested, please see page 8, lines 256 to 265

Paragraph 5 – fine, no comments

Authors’ response:

Okay

Paragraph 6 – Systematic review on acceptance, interesting but need to mention acceptance in results Suggest starting with what the study found (to put it in context), then introduce the systematic review and corroborating findings

Authors’ response:

We have included information on acceptance as requested in the results. Please see new paragraph summarizing the acceptance rates on page 10, lines 310 to 321 of the result section.

Paragraph 7 – fine, no comments

Authors’ response:

Okay

Paragraph 8 – PEPFAR/inpatient ward– suggest weaving into primary findings, since it is provides support for inpatient findings.

Authors’ response:

Now integrated into a new paragraph 4 in the revised manuscript, please see page 4, lines 281 to 294 in the discussion section of the revised manuscript

Paragraph 9 – fine, no comments

Conclusion

Question about this sentence: “Specifically, these modalities identified high numbers of children living with HIV while maintaining high yields”

isn’t identifying high numbers of children living with HIV yield? Please clarify.

Authors’ response:

We have effected change. The sentence now reads as follows: “Specifically, these modalities identified high numbers of children living with HIV while maintaining high HIV positivity rates”

Please see conclusion on page 12, lines 383 to 384 of marked up version of revised manuscript.

Reviewer 3 Comments and Authors’ responses:

Title: In the title, what is the meaning of “high-yield facility based settings’’? Did you mean “ high-yield point of service?’’ Somehow, the current title is not clear. 

Authors’ response:

Thank you for your comment. Authors agree with suggestions. The Title of the manuscripts now reads as follows: “Increasing pediatric HIV testing positivity rates through focused testing in high-yield point of service in health facilities—Nigeria, 2016-2017”. Please review the title page of marked up version of revised manuscript. On page 1.

Abstract

1. Reading the background, the problem you intend to solve is increasing pediatric ART in Nigeria. Thus, can you consider including the contribution of each POS to HIV seropositivity. This is important to show the contribution of POPD. Actually, though POPD has a lower yield, it has a meaningful contribution in case finding. 

Authors’ response:

We have now included the contributions of the various POS to the total HIV positive children found in the abstract of the revised manuscript. The following excerpt was included in abstract of revised manuscript: “The percentage contribution to total HIV positive children found by point of services was: Family index modality= 55.8% (95%CI: 40.9-69.8%); POPD=20.9% (95%CI: 11.3-35.6%); inpatient ward=20.9 (95%CI: 11.3-35.6%) and TB Clinic=2.3% (95%CI: 0.3-14.8%).”

Please see page 1, lines 38 to 40 of marked up version of revised manuscript.

2. In the sentence “….group were 1-4 years =752 (41.3%); 10-14 years= 320 (17.6%);

5-9 years= 520 (28.5%); and <1 years=230 (12.6%)’’, the ordering of the age group is difficult to follow. I would order the age groups from the youngest to the oldest. 

Authors’ response:

We have now ordered age group by youngest to oldest. See excerpt included in abstract of revised manuscript: “The numbers of children tested by age group were <1 years=230 (12.6%); 1-4 years =752 (41.3%); 5-9 years= 520 (28.5%); and 10-14 years= 320 (17.6%).” 

Please see page 1, lines 31 to 33 of marked up version of revised manuscript.

3. The conclusion does not tie with the results. TB clinic has the highest yield, therefore should come first in order of priority, follow by pediatric inpatient wards and family index testing. 

Authors’ response:

In this instance, the ordering in the conclusion was correct. Although TB clinic had the highest odds (TB clinic=7.2 (95% CI: 0.9-60.9)) for finding an HIV positive child. A careful examination of the confidence interval (CI) show a very wide CI starting from less than 1. This was because of the relative small number of HIV positive children identified at TB Clinic (2.3% of total positive children) compared with other POS. However, we have revised language to show the importance of family index texting. Please see page 1, lines 45 to 47 of marked up version of revised manuscript.

Introduction 

General comments: 

The problem statement (challenges in improving HIV case finding among children) and the efforts deploy by the Government of Nigeria (GON) and partners (notably PEPFAR) to address the issue is well described. However, there is no mentioned of the outcome implementation of the Nigeria National Acceleration Plan for Pediatric HIV treatment and Care, 2016-2018. This is the national document that introduced family index testing as a key strategy for pediatric HIV case finding in Nigeria. Thus, making reference to this Plan should further highlight the importance of the topic as well as the relevance/significance of your study. Dr. Uzoma Ene (co-author of this paper) was among the experts who elaborated this Acceleration Plan. I believe she can contribute in addressing this point. 

Authors’ response:

We have included a reference of the National Acceleration Plan for Pediatric HIV treatment and Care, 2016-2018 in the introduction. Please see page 1, lines 63 to 66 of marked up version of revised manuscript.

Specific comments:

Background: 

Line 4-5: I would suggest updating the figure with current UNAIDS statistics (https://www.unaids.org/en/regionscountries/countries/nigeria)

i) 220 000 is the upper limit of the CI. I would use but the point estimate which is 140 000 and indicate the CI= [91 000 - 220 000] 

Authors’ response:

The information earlier included were those available at the time of development of manuscript. We have now updated with current data. Please review lines 50 to 54 of the revised manuscript

ii) 26% ART coverage is not current. Nigeria current ART coverage for children 0-14 is 35% [22 - 53].

Authors’ response:

We have now updated with current data. Please review page 2, lines 50 to 56 of the marked-up version of the revised manuscript

iii) References [3] and [4] are the same (https://www.unaids.org/en/regionscountries/countries/nigeria). Please, delete one. 

Authors’ response:

We have identified the duplicate reference and deleted as appropriate. Please review lines 50 to 54 of the marked-up version of the revised manuscript

Second to the last sentence of the 2nd paragraph: 

Reference #9: please update with 2018 data: ART coverage: males= 37%; females=68% [https://aidsinfo.unaids.org].

Authors’ response:

We have now updated with current data. Please review lines 66 to 74 of the marked-up version of the revised manuscript

Last sentence of the 1st paragraph: ‘’Major reasons for the disproportionately lower coverage among children are lower HIV testing positivity rates, indicating poorly targeted testing, and lower linkage to treatment among children’’: I think lower HIV testing positivity rate is just one of the reasons for the current low pediatric ART coverage. I would revise this sentence to indicate that major reasons for the disproportionately lower coverage among children include among others lower HIV testing positivity rates.

Authors’ response:

Authors have revised sentence as recommended, please see page 3, lines 72 to 74 of the marked-up version of the revised manuscript

Methods

Strategies implemented during PICF pilot

P.4-Please describe how the index testing modality was implemented. Is it like outpatient parents diagnosed HIV positive were invited to bring their children for testing? or were index cases recruited among parents in HIV services in hospital (e.g: ART clinic or PMTCT)? A thorough description of the implementation of the index case testing is needed to understand the processes involved and how children were recruited for testing during the intervention period. 

Authors’ response:

We have included a detailed description of the family index testing modality in the method section. See excerpt included in section on strategies implemented during PICF pilot of the marked-up version of the revised manuscript (Please see page 5, lines 138 to 144): The family index modality identified new cases through a previously identified case, i.e. children of an HIV positive mother or siblings of an index child. HIV positive mothers were identified during posttest counselling, at the ART clinic, ante-natal clinic, post-natal clinic or outpatient clinics and encouraged to return to clinic with their children after counselling for a free HIV test. Additionally, parents of a previously identified HIV positive child were also encouraged to bring other children with unknown HIV status for an HIV test.

P.5-3rd sentence: ‘’Additionally, there was 100% index testing of children with HIV-positive mother or siblings’’: This sentence is confusing when reading table 2 which indicates that 85.9% of children were successful tested through family index modality. How should we understand this difference? Moreover, does it mean only children of HIV-positive mothers were tested? How about fathers who tested positive in the hospital? Were they not also ask to bring their children for testing? If no, why? What was really the entry point for family index case testing? OPD, PMTCT, ART clinic?

Authors’ response:

We have included a detailed description of the family index testing modality in the method section. See revised manuscript (lines122 to 127). The previously confusing sentence ‘’Additionally, there was 100% index testing of children with HIV-positive mother or siblings’’ has been deleted. This pilot didn’t not directly focus on fathers, because the majority of HIV children are infected by vertical transmission from an HIV positive mother to child. The entry points for family index has been described, please see lines 122 to 127.

P.5- ‘’These PICF strategies were piloted at the TB clinics, pediatric inpatient medical wards,

adult outpatient clinic among families of identified index cases, POPDs, and the immunization clinics in the selected sites’’: where there ART and malnutrition clinics in the selected health facilities? if yes, why were they not included in the study knowing that these are high yield HIV case finding point of services?

Authors’ response:

The clinics where the study was conducted were secondary level hospitals, the malnutrition clinics were not clearly defined. Most operated as part of the OPD. Furthermore, ART was not a testing stream for this study, only HIV positive children were seen at ART. However, the ART clinic served as a source for identifying children with siblings with unknown status that formed part of the family index testing modality.

Results

 P.6. Last paragraph: Age and POS were the only predictors of seropositivity assessed (Table 3). Is it possible to assess more children-level socio-demographic factors such as sex, educational level among others?

Authors’ response:

Unfortunately, we couldn’t include other socio-demographic factors in the analysis model because they were not available. Our study is a secondary analysis of data obtained from a pilot. We are limited by available data.

Table 3: I think the title should read ‘’Unadjusted and adjusted odds ratios of pediatric HIV case detection by patient age group and POS''

Authors’ response:

Authors agree with reviewer’s suggestions. We have made the suggested changes to table 3

Discussion

2nd sentence-1st paragraph: ‘’Several program strategies for improving efficiency of testing through increasing HIV positivity yield, while maintaining high numbers of HIV positive children found, and for improving efficacy of testing by assuring linkage to ART were implemented in selected health facilities providing HIV care’’.---� This sentence is too long and difficult to understand. What is the message here?

Authors’ response:

This sentence have been deleted

2nd sentence-2nd paragraph: ‘’This may be because of a continued program emphasis on PMTCT: perhaps these youngest children were less likely to have unknown status or test positive because their mothers’ status had been confirmed during pregnancy’’---� the explanation of the lowest proportion of children with unknown HIV status and tested HIV positive should be reviewed. Actually, youngest children were less likely to have unknown HIV status and test HIV positive not because ‘’ their mothers’ status had been confirmed during pregnancy’’ , but more likely because these youngest children may have been tested for HIV and those tested HIV+ received ARV prophylaxis and this through the PMTCT program. This is suggestive of the effectiveness of PMTCT program in the selected health facilities. 

Authors’ response:

We have provided more clarification to the sentence in the revised manuscript. However, this paragraph have been moved down as requested by another reviewer. Please see page 11, lines 361 to 373 of the marked up version of the revised manuscript.

4th sentence-3rdparagraph: ‘’Even with the use of the HIV risk screening tool in the POPD, this POS gave a low yield of 1.0%’’. ---� This is in my opinion a key finding of this paper considering the potential implications. Actually, the aim of the risk screening tool was to improve the yield. If this yield is still low at 1%, therefore the tool is not achieving the desired results. The reported yield in your study is even lower compared to the yield of the DHT (diagnostic HIV testing=symptoms-based testing strategy) among children reported at 4% in Cameroon (https://journals.plos.org/plosone/article/comments?id=10.1371/journal.pone.0214251). This suggests that compared to the Bandason screening tool, better yield can even be achieved focusing on testing only HIV symptomatic outpatient children. There is urgent need to re-assess the validity of the Bandason screening tool in low HIV prevalence context such as Nigeria. 

Authors’ response:

We agree with reviewers comment, we have included that perspective in the revised manuscript. Please lines 281 to 294 of the marked up version of the revised manuscript.

P.8: 1st sentence: ‘’Children for HIV testing (family index testing), found that the rate of pediatric case identification increased 3.8-fold from 3.5% to 13.6% [15]’’---� This statement is wrong because the reported figure correspond to the increase in HIV testing uptake and not case identification rate that was 7.4%. Please check your reference again. 

Authors’ response:

We have revised sentence to make it more explicit. Please see lines 256 to 259 of the marked up version of the revised manuscript.

3rd sentence: ‘’Our study results were comparable with findings in these studies’’. --�Which of your study results are you referring to? In your study, you did not compare the HIV testing uptake (before and after the intervention) instead you reported the family index case testing HIV positivity rate of 3.5% which is far different from the 7.4% reported by Wagner et al. in Kenya. Instead, your HIV positivity rate (3.5%) corresponds exactly to the 3.5% reported in the family index case testing study in the neighbouring Cameroon by Yumo et al. (see reference#10 of your manuscript). 

Authors’ response:

We have provided a clearer description in the revised manuscript. Please see lines 266 to 273 of the marked up version of the revised manuscript.

2nd sentence-3rd paragraph: ‘’With exception of the immunization clinic, our study found a

much higher acceptance rate among all the provider-initiated testing and counseling POS in our study sites, which may indicate that the pre-test counseling being done for families of at-risk pediatric patients at these sites is effective’’---� You may support your findings with evidence from neighbouring Cameroon equally showing higher acceptance (99.7%) among parents (Yumo et al.=> see ref 10) and Ida Penda et al( .https://www.ncbi.nlm.nih.gov/pmc/articles/PMC6090739/). 

Authors’ response:

We have included additional comparison with studies from Yumo et al. Please see lines 310 to 321 of revised manuscript.

2nd sentence- 4th paragraph: ‘’About 24% of children with unknown status identified through the family index POS were not tested for HIV’’----�This is about 2 times less than the 44% reported by Yumo et al. in Cameroon (see reference 10 of your manuscript). Can you explain this better outcome?

Authors’ response:

We will not be able to explain the differences in outcomes in the two studies. However, we have compared findings in both studies to enable readers have both perspective. Please review description in lines 335 to 344 of the marked up version of the revised manuscript.

4th sentence-4th paragraph: ‘’Our study did not investigate the reasons for the failure of parents or guardians to return with their children’’.---�These reasons were investigated by Yumo et al. in a similar study in Cameroon (see reasons in ref. above). 

Authors’ response:

We have reviewed reports from Yumo et al, their insinuation was similar with what was already described in our manuscript. Please review description in lines 335 to 351 of the marked up version of the revised manuscript.

2nd sentence-5th paragraph: ‘’Our findings that pediatric inpatient ward was an efficient POS for finding children living with HIV while maintaining a yield higher than the overall yield corroborates that guidance’’--�However, the contribution of inpatient (20.9%) to case finding load is similar to POPD (20.9%) and far lower to family index testing modality (55.8%). Thus, there is need for the combination of these POS, even if it means somehow losing in efficiency at the POPD. This finding simply shows the challenge in achieving optimal coverage while maintaining efficiency. 

Authors’ response:

We agree with reviewers comment. Actually our follow on sentence in same paragraph (lines 307 to 312) makes that point exactly. Although, the contribution of inpatient (20.9%) to case finding load was similar to POPD (20.9%), a lot more test was done at POPD to find the same number of patients. Our recommendation to use risk stratification will ensure programs test the most at risk population and improve testing efficiency at POPD.

3rd sentence-5th paragraph: ‘’ We found that the family index testing modality was the most efficient POS for all sites, accounting for about 38% of total children tested but contributing more than half (55.8%) of all the positives identified’’--�This was the key strategy outlined in the National Acceleration Plan for Pediatric HIV Treatment and Care for Nigeria (Please, see Dr Uzoma Ene for reference). This finding indicates that if implemented at scale, this Plan could help in bridging the current gap in pediatric ART in Nigeria. 

Authors’ response:

We agree with reviewers comment. We have made that point in the revised manuscript. Please see page 9, lines 281 to 294, in discussion of marked up version of revised manuscript.

3rd sentence-5th paragraph: ‘’While we found a low yield in the POPD, because so many children seek care there in Nigeria, testing there may still represent a way to find more children living with HIV’’-� This is correct, but contradicts a little bit your statement above re prioritizing inpatients testing.

Authors’ response:

We have rephrased sentence to make it clearer. Please see lines 291 to 294 under discussion in the revised manuscript.

4th sentence-5th paragraph: ‘’ However, in order for this to be efficient, there needs to be a program emphasis on fidelity to the HIV risk stratification tool’’-� This sentence is not clear. Are you referring here to the need of using a screening tool? if yes, as indicated above, please refer to comment above re the use of the Bandason screening tool in Nigeria. 

Authors’ response:

We have rephrased sentence to make it clearer. Please see page 9, lines 291 to 294 under discussion in the revised manuscript.

2nd sentence-6th paragraph: ‘’However, the purposive nature of site selection limits the generability of the findings from our analysis’’-� Other high yield POS such ART clinics (for index case testing) and malnutrition clinics were not explored. This is another limitation of the study because information from these POS could have given more beef to the paper.

Authors’ response:

The ART clinic was one of the places used to elicit index for testing. Please refer to lines 138 to 144 of the revised manuscript. However, we have included our inability to thoroughly explore malnutrition clinics as a limitation. Please see lines 374 to 379 under discussion in the revised manuscript.

Conclusion

3rd sentence: ‘’The study findings suggested that to improve facility-based HIV positivity rates

among children aged 0–14 years in Nigeria, an increased focus on HIV testing among children seeking care in pediatric inpatient wards, through family index testing, and perhaps in TB clinics is appropriate’-� in this sentence, I would remove ''perhaps'', include the POPD as another priority testing point while recommending the re-assessment of the validity of the Bandason tool in low HIV prevalence context such as Nigeria. We can't afford not to test children at the OPD as it’s an important POS considering it contribution in pediatric HIV case finding load. 

Authors’ response:

We have rephrased the conclusion in line with suggestions from reviewer. Please see lines 370 to 379 in revised manuscript. We have also made the point for the need to re-assess the validity of the Bandason tool in low HIV prevalence context such as Nigeria in lines 279 to 287 of revised manuscript. However, we could not remove the word ‘perhaps’ from our description for TB clinic becaue our study was not sufficiently powered to give an accurate inference for testing in the TB clinic.

Others: The review of this paper was made unmercenary difficult because the lines are not numbered. Please, kindly numbered the lines in the revised version before re-submission.

Authors’ response:

Manuscript text are now numbered.

 

References

1. Martinez-Ezquerro JD, Riojas-Garza A, Rendon-Macias ME. [Clinical significance vs statistical significance. How to interpret the confidence interval at 95]. Rev Alerg Mex. 2017;64(4):477-86. Epub 2017/12/19. doi: 10.29262/ram.v64i4.334. PubMed PMID: 29249109.

---

## [Decision Letter · Decision Letter 1]

16 Mar 2020

PONE-D-19-27505R1

Increasing pediatric HIV testing positivity rates through focused testing in high-yield points of service in health facilities—Nigeria, 2016-2017

PLOS ONE

Dear Dr. Odafe,

Thank you for submitting your manuscript to PLOS ONE. After careful consideration, we feel that it has merit but does not fully meet PLOS ONE’s publication criteria as it currently stands. Therefore, we invite you to submit a revised version of the manuscript that addresses the points raised during the review process.

We would appreciate receiving your revised manuscript by Apr 30 2020 11:59PM. To enhance the reproducibility of your results, we recommend that if applicable you deposit your laboratory protocols in protocols.io, where a protocol can be assigned its own identifier (DOI) such that it can be cited independently in the future. For instructions see: http://journals.plos.org/plosone/s/submission-guidelines#loc-laboratory-protocols

We look forward to receiving your revised manuscript.

Kind regards,

Joseph Fokam, Ph.D

Academic Editor

PLOS ONE

Reviewers' comments:

Reviewer's Responses to Questions

**Comments to the Author**

1. If the authors have adequately addressed your comments raised in a previous round of review and you feel that this manuscript is now acceptable for publication, you may indicate that here to bypass the “Comments to the Author” section, enter your conflict of interest statement in the “Confidential to Editor” section, and submit your "Accept" recommendation.

Reviewer #3: All comments have been addressed

2. Is the manuscript technically sound, and do the data support the conclusions?

Reviewer #3: Yes

3. Has the statistical analysis been performed appropriately and rigorously? 

Reviewer #3: Yes

4. Have the authors made all data underlying the findings in their manuscript fully available?

Reviewer #3: Yes

5. Is the manuscript presented in an intelligible fashion and written in standard English?

Reviewer #3: Yes

6. Review Comments to the Author

Reviewer #3: The authors have adequately addressed my comments raised in a previous version but for reference #10 which is not well cited in line 288. Here is the correct reference: https://doi.org/10.1371/journal.pone.0214251. Please, note that even though this article is from the same author (Yumo et al.), it’s different from reference # 10 well cited for example in line 271. Please, amend this reference number and the references list as well.

7. PLOS authors have the option to publish the peer review history of their article (what does this mean?). If published, this will include your full peer review and any attached files.

Reviewer #3: Yes: Habakkuk A. Yumo

---

## [Author Response · Author response to Decision Letter 1]

17 Mar 2020

Reviewer #3: 

The authors have adequately addressed my comments raised in a previous version but for reference #10 which is not well cited in line 288. Here is the correct reference: https://doi.org/10.1371/journal.pone.0214251. Please, note that even though this article is from the same author (Yumo et al.), it’s different from reference # 10 well cited for example in line 271. Please, amend this reference number and the references list as well.

Authors’ response:

We have made the required changes to the reference. Please review the lines 298 and 442 to 445 of the Tracked version of the revised manuscript. The journal is now correctly cited as reference #30 in line 298 and correctly referenced in lines 442 to 445.

---

## [Decision Letter · Decision Letter 2]

2 Jun 2020

Increasing pediatric HIV testing positivity rates through focused testing in high-yield points of service in health facilities—Nigeria, 2016-2017

PONE-D-19-27505R2

Dear Dr. Odafe,

We are pleased to inform you that your manuscript has been judged scientifically suitable for publication and will be formally accepted for publication once it complies with all outstanding technical requirements.

With kind regards,

Jose A. Bauermeister, MPH, PhD

Academic Editor

PLOS ONE

Additional Editor Comments (optional):

Dear authors,

Thank you for your careful revision of your manuscript. Please note that the Reviewer noted a change must be addressed in your referencing of the following sentence:

"However, the percentage of children with unknown status that was not tested (24%) in our study was much lower than the 43.3% reported by Yumo et al for neighboring Cameroon[30]".

The Reviewer noted that tis reference corresponds to number 10 in your current reference list, instead of 30. Please revise accordingly.

Reviewers' comments:

Reviewer's Responses to Questions

**Comments to the Author**

1. If the authors have adequately addressed your comments raised in a previous round of review and you feel that this manuscript is now acceptable for publication, you may indicate that here to bypass the “Comments to the Author” section, enter your conflict of interest statement in the “Confidential to Editor” section, and submit your "Accept" recommendation.

Reviewer #3: (No Response)

2. Is the manuscript technically sound, and do the data support the conclusions?

Reviewer #3: Yes

3. Has the statistical analysis been performed appropriately and rigorously? 

Reviewer #3: Yes

4. Have the authors made all data underlying the findings in their manuscript fully available?

Reviewer #3: Yes

5. Is the manuscript presented in an intelligible fashion and written in standard English?

Reviewer #3: Yes

6. Review Comments to the Author

Reviewer #3: References (10 and 30) are still inadequately cited. Please see my comments in the attached manuscript.

7. PLOS authors have the option to publish the peer review history of their article (what does this mean?). If published, this will include your full peer review and any attached files.

Reviewer #3: Yes: Habakkuk A. Yumo

---

## [Editor Report · Acceptance letter]

10 Jun 2020

PONE-D-19-27505R2 

 Increasing pediatric HIV testing positivity rates through focused testing in high-yield points of service in health facilities—Nigeria, 2016-2017 

Dear Dr. Odafe:

I'm pleased to inform you that your manuscript has been deemed suitable for publication in PLOS ONE. Congratulations! Your manuscript is now with our production department. 

Kind regards, 

on behalf of

Dr. Jose A. Bauermeister 

Academic Editor

PLOS ONE